# Direct Interaction of Mitochondrial Cytochrome *c* Oxidase with Thyroid Hormones: Evidence for Two Binding Sites

**DOI:** 10.3390/cells11050908

**Published:** 2022-03-06

**Authors:** Ilya P. Oleynikov, Roman V. Sudakov, Natalia V. Azarkina, Tatiana V. Vygodina

**Affiliations:** 1A.N. Belozersky Institute of Physico-Chemical Biology, M.V. Lomonosov Moscow State University, Leninskie Gory 1, Bld. 40, 119992 Moscow, Russia; oleynikov.biophys@gmail.com (I.P.O.); vygodina@belozersky.msu.ru (T.V.V.); 2N.N. Blokhin Russian Cancer Research Center, Kashirskoye Shosse 24, 115478 Moscow, Russia; sudakovromvlad@gmail.com

**Keywords:** cytochrome oxidase, thyroid hormones, steroid hormones, bile acid-binding site, superoxide generation, regulation

## Abstract

Thyroid hormones regulate tissue metabolism to establish an energy balance in the cell, in particular, by affecting oxidative phosphorylation. Their long-term impact is mainly associated with changes in gene expression, while the short-term effects may differ in their mechanisms. Our work was devoted to studying the short-term effects of hormones T2, T3 and T4 on mitochondrial cytochrome *c* oxidase (CcO) mediated by direct contact with the enzyme. The data obtained indicate the existence of two separate sites of CcO interaction with thyroid hormones, differing in their location, affinity and specificity to hormone binding. First, we show that T3 and T4 but not T2 inhibit the oxidase activity of CcO in solution and on membrane preparations with *K*_i_ ≈ 100–200 μM. In solution, T3 and T4 compete in a 1:1 ratio with the detergent dodecyl-maltoside to bind to the enzyme. The peroxidase and catalase partial activities of CcO are not sensitive to hormones, but electron transfer from heme *a* to the oxidized binuclear center is affected. We believe that T3 and T4 could be ligands of the bile acid-binding site found in the 3D structure of CcO by Ferguson-Miller’s group, and hormone-induced inhibition is associated with dysfunction of the K-proton channel. A possible role of this interaction in the physiological regulation of the enzyme is discussed. Second, we find that T2, T3, and T4 inhibit superoxide generation by oxidized CcO in the presence of excess H_2_O_2_. Inhibition is characterized by *K*_i_ values of 0.3–5 μM and apparently affects the formation of O_2_^●−^ at the protein surface. The second binding site for thyroid hormones presumably coincides with the point of tight T2 binding on the Va subunit described in the literature.

## 1. Introduction

Cytochrome *c* oxidase (CcO) (for recent reviews, see [1,2]) is the terminal enzyme of the respiratory chain located in the coupling membranes of mitochondria and aerobic bacteria. CcO catalyzes the reduction of oxygen to water, providing the cell with the energy released. The catalytic center of the enzyme is a binuclear center *a_3_*/Cu_B_ formed by closely spaced high-spin heme *a_3_* and copper ion. Electrons arriving from cytochrome *c* are transferred along the redox centers: Cu_A_ → low-spin heme *a* → *a_3_*/Cu_B_ → O_2_. The four-electron reduction of oxygen to water in the binuclear center is coupled to proton translocation and energy transduction to ∆μH^+^. All protons involved in the catalysis are transferred inside the protein via K- and D-proton channels. Mitochondrial CcO consists of 13 subunits [3], the three largest of which (catalytic subunit I containing heme *a*, *a_3_*/Cu_B_ binuclear center and the proton channels, subunit II carrying cytochrome *c* binding site and Cu_A_ and then subunit III) are homologous to the subunits of bacterial enzymes and encoded by mitochondrial DNA. The role of the smaller subunits encoded by the nuclear genome is not clear and presumed to be related to enzyme regulation [4]. The catalytic cycle of CcO is subdivided into two phases: eu-oxidase and peroxidase [5,6]. It is accepted that in the eu-oxidase phase, two “chemical” protons are delivered to the binuclear center via the K-channel. In the peroxidase phase, the K-channel is closed and protons are transferred across the membrane through channel D (for details, see reviews [6,7,8]).

New budding perspectives for studies on CcO regulation have arisen from the series of works from Ferguson-Miller’s laboratory. The authors revealed a conserved hydrophobic cavity in the crystal structure of CcO from *Rhodobacter sphaeroides* and bovine heart mitochondria, and suggested that it could represent a binding site for small amphipathic molecules [9,10,11,12,13]. The cavity is located on the inner side of the membrane at the boundary of subunits I and II and near the mouth of proton channel K. On the X-ray structures of CcO from mitochondria and *R. sphaeroides*, it was occupied by the bile acid molecule (cholate or deoxycholate, respectively) used in crystallization so the cavity was called the bile acid-binding site (BABS). In the structure of a bacterial enzyme, a decyl-maltoside detergent molecule was also visualized, located in the immediate vicinity of BABS and thought to reach it in some conformations [9,11]. In the structure of eukaryotic CcO, a similar place is occupied by a tightly bound phospholipid molecule [9]. Computational analysis of the BABS structure reveals its possible affinity to a number of physiologically active molecules including steroids, triiodide-thyronine, nicotinamide, flavin, some nucleotides and products of heme decomposition [11,12]. In some cases, and in particular, for triiodide-thyronine, the inhibition of CcO from *R. sphaeroides* was shown experimentally [12]. At the same time, no effect of steroid sex hormones on the activity of the bacterial enzyme was found. However, our group showed recently that steroid hormones, as well as secosteroids and detergent Triton X-100 (which mimic estrogen in some respects), are able to bind with BABS and inhibit the activity of mitochondrial CcO [14,15].

It seems that BABS has fairly broad specificity and is able to interact with a variety of physiologically active molecules of amphipathic structure. Of particular interest is an inhibitory effect of the thyroid hormone T3 on the activity of wild-type CcO from *R. sphaeroides* (*K*_i_ in the range of 30–210 μM, depending on the conditions of the assay) [12]. Obviously, triiodide-thyronine cannot be considered as a physiological regulator of a bacterial enzyme, but in the case of a eukaryotic analog, its regulatory function looks quite probable.

Thyroid and steroid hormones are metabolic regulators of the highest rank and their signaling pathways intersect in many cases [16,17,18,19]. The ability of such regulatory molecules to influence the activity of CcO, a key enzyme of aerobic metabolism, through direct interaction is quite unexpected. Investigation of this phenomenon can be important from different points of view: in the physiological aspect, for a better understanding of the CcO catalytic mechanism and finally, in terms of the evolution of regulatory systems. The aim of the present work was to study the effects of the “classic” thyroid hormones T3 (L-3,5,3′-triiodide-thyronine) and T4 (thyroxine, or L-3,5,3′,5′tetraiodo-thyronine), as well as T2 (3,5-diiodo-thyronine), the most active product of T3 deiodination, on the activities of mitochondrial CcO in solution and membrane preparations. The results obtained suggest that the mitochondrial enzyme contains two different sites of interaction with thyroid hormones.

## 2. Materials and Methods

*Chemicals.* Thyroxine (T4), 3,5,3′ triiodo-thyronine (T3), 3,5 diiodo-thyronine (T2), sodium dithionite, cholate, succinate, ATP, cytochrome *c* (type III from an equine heart), TMPD (N,N,N′,N′-tetramethyl-*p*-phenylenediamine), hexaammineruthenium (III), L-ascorbic acid, potassium ferro- and ferricyanide, asolectin from soya beans (type II S), superoxide dismutase from bovine erythrocytes, xanthine oxidase from bovine milk, hypoxanthine and hydrogen peroxide were sourced from Sigma-Aldrich (Saint Louis, MO, USA). pH-buffers, MgSO_4_, sucrose and EDTA (ethylenediaminetetraacetic acid) were sourced from Amresco (Solon, OH, USA). Water-soluble tetrazolium (WST-1) was sourced from Dojindo Molecular Technologies (Kumamoto, Japan). Bio-Beads SM-2 Adsorbent 20–50 mesh was sourced from Bio-Rad (Hercules, CA, USA). Dodecyl-maltoside of the “Sol-Grade” type was sourced from Anatrace (Maumee, OH, USA).

Thyroid hormones T4, T3 and T2 were dissolved in 0.1 M NaOH.

Hydrogen peroxide solution (about 30%) was kept at 4 °C. Before the experiments, the concentration was checked spectrophotometrically using molar extinction coefficient ε_240_ = 40 M^−1^·cm^−1^ [20].

Marketable adsorbent balls Bio-Beads were activated by washing three times with methanol (10g of balls were stirred for 20 min with 60 mL methanol), followed by a slow rinse with water (*c.a.* 700 mL) on a funnel. Activated balls were kept covered with water in a refrigerator.

*Preparations. Cytochrome c oxidase* (CcO) was purified from heart mitochondria of *Bos taurus*. Hearts were purchased from the abattoir of Pushkinsky Myasnoy Dvor Ltd. (Pushkino, Moscow region, Russia) and stored on ice for 2–3 h after slaughter until the isolation procedure began. CcO was isolated according to a modified method of Fowler et al. [21] described previously [22]. The concentration was determined from the difference absorption spectra (dithionite reduced vs. air oxidized) using molar extinction coefficient ∆ε_605–630nm_ = 27 mM^−1^·cm^−1^. CcO destined for reconstitution into proteoliposomes was preliminarily purified on a sucrose gradient as described in [23].

*CcO reconstruction into phospholipid vesicles*. Asolectin was dispersed to a concentration of 60 mg/mL in 50 mM potassium phosphate buffer (pH 7.6) supplemented with 2 mM MgSO_4_ and 1.3% cholate. The mixture was bubbled with argon and then sonicated five times for 30 s in an ultrasonic disintegrator (Branson Sonifier 150, Brookfield, CT, USA). The residual nonsonicated material and metal concomitants from the pestle were removed by centrifugation for 5 min at 14,000 rpm in a mini-centrifuge (Eppendorf, Hamburg, Germany). CcO was added to a clarified lipid solution up to 1 µM and stirred for 30 min. Liposomes with CcO were formed during the process of detergent removal, achieved by vigorous stirring of the sample with Bio-Beads balls. The balls were added at room temperature, sequentially: 80 mg/mL of suspension (30 min stirring), 80 mg more (60 min stirring), 160 mg more (120 min stirring) and 160 mg more (120 min stirring). Proteoliposomes were collected and dialyzed overnight against 50 mM potassium phosphate buffer pH 7.6 supplied with 2 mM MgSO_4_, at +4 °C. The orientation of CcO in the membrane vesicles was assessed by comparing the heme *a* reduction levels achieved in the presence of 5 mM ascorbate + 2.5 μM cytochrome *c* alone, or with 0.1 mM TMPD as a membrane-penetrating redox mediator.

*Isolation of rat liver mitochondria.* Rat liver mitochondria were isolated by differential centrifugation as described in [24] in a medium containing 250 mM sucrose, 5 mM MOPS, 1 mM EGTA and bovine serum albumin (0.5 mg/mL), pH 7.4. The final washing was performed in a medium of the same composition. The protein concentration was determined by the Biuret method. The experimental medium contained 250 mM sucrose, 5 mM MOPS and 1 mM EGTA, pH 7.4. Mitochondria were kept frozen at −20 °C and thawed just before the experiment. A single freeze-thaw procedure resulted in destruction of the outer mitochondrial membrane.

*Preparation of bovine heart submitochondrial particles (SMP).* The heavy fraction of bovine heart mitochondria was suspended in 40 mL of 0.3 M sucrose supplemented with 5 mM MgSO_4_, 1 mM succinate, 1 mM ATP and 10 mM Hepes, pH 8.0. The mixture was bubbled with argon and then sonicated two times for 55 s each in an ultrasonic disintegrator, at 44 kHz, 0.4 A. After sonication, the pH of the mixture was adjusted to 8.0 by Tris powder. Mitochondrial debris was removed by centrifugation for 12 min at 15,000 rpm (Beckman JA-20, Brea, CA, USA). SMP were precipitated from supernatant at 100,000 g for 1 h, washed by 0.3 M sucrose with 10 mM Hepes, pH 7.5 and precipitated as previously described. The final precipitate was suspended in a small volume of the same buffer and frozen [25]. To determine the CcO concentration in the sample, a small amount of SMP was solubilized in 50 mM Hepes/Tris buffer, pH 7.6 containing 1% DM.

*Spectroscopic assays* were carried out in a standard semi-microcuvette (Hellma, Mullheim, Germany) with blackened side walls and a 10-mm light pathway.

*Absolute absorption spectra* were recorded with a 3 nm-slit width, at a speed of 2 nm/s, on a double-beam spectrophotometer Cary 300 Bio (Varian, Palo Alto, CA, USA).

*The kinetics of spectral changes* were monitored on spectrophotometer SLM Aminco DW-2000 (SLM Instruments, Urbana, IL, USA) in dual-wavelength mode.

*Stopped-flow experiments* were performed using a stopped-flow spectrophotometer (Applied Photophysics SX-20, Leatherhead, UK) operated in the diode array mode, using a 20-μL cell with a 1-cm optical pathway. The spectra in the 280–720 nm range were collected with a minimal interval of 1 ms.

*Measurements of enzymatic activities*. The CcO activity was monitored using two different methods.

*The rate of oxygen consumption* was measured with a covered Clark-type electrode using an Oxygraph-type device (“Oxytherm” from Hansatech, Pentney, UK), in a thermostatted cell at 25 °C with permanent stirring. Then, 5 mM ascorbate, 0.1 mM TMPD and 10 μM cytochrome *c* were used together as an oxidation substrate. The concentration of CcO in the experimental medium varied from 18 to 26 nM. The assays on solubilized CcO were performed in the basic medium containing 50 mM Hepes/Tris buffer, pH 7.6, 0.1 mM EDTA, 50 mM KCl and 0.05% dodecyl-maltoside (DM) to maintain the enzyme in the solubilized state. In some experiments, the DM concentration varied from 0.02% to 1% as indicated in the legends to the figures. Concentrations of DM are given throughout the text both in % and mM (0.05% ≈ 1.0 mM DM). The assays on proteoliposomes were carried out in 50 mM potassium-phosphate buffer, pH 7.6 supplemented with 0.1 mM EDTA and 1 µM CCCP. Respiration of whole mitochondria and SMP was registered in 50 mM Hepes/Tris, pH 7.6 supplemented with 0.3 M sucrose, 0.5 mM EDTA and 0.5 μM CCCP. The turnover (TN) values of CcO refer to electrons transferred in 1 s per enzyme monomer.

*The rate of cytochrome c^2+^ oxidation* was followed spectrophotometrically as the absorption difference at 550 nm vs. the 535 nm reference in a dual-wavelength mode. The initial linear part of the traces was used for calculations. Cytochrome *c* was reduced beforehand by dithionite. The rest of the dithionite was removed on a Sephadex G-25 coarse column.

*The peroxidase activity of CcO* was assayed in the basic medium, pH 7.6 following spectrophotometric peroxidation of ferrocyanide by the absorption difference at 420 nm vs. the 500 nm reference. To obtain a redox buffer with E_m_ ≈ 420 mV, equal amounts of ferri- and ferrocyanide were mixed. To facilitate the interaction of ferrocyanide with CcO, 0.2 μM cytochrome *c* was present. No ferrocyanide oxidase activity was observed until the peroxidase reaction was initiated by the addition of H_2_O_2_.

*The catalase activity of CcO* was detected by a Clark-type electrode in the presence of excess hydrogen peroxide, in 30 mM potassium-phosphate buffer, pH 7.1 supplemented with 2 mM EDTA and 0.05% DM, as described in [26]. The observed oxygen release was completely sensitive to cyanide.

*The reaction of superoxide generation* catalyzed by CcO in the presence of excess hydrogen peroxide was assayed in the basic medium, pH 7.6 by the classic method [27], monitoring the superoxide dismutase-sensitive reduction of tetrazolium salts to formazan. The water-soluble tetrazolium dye (WST-1) reduction was followed as the absorption difference at 442 nm vs. the 552 nm reference.

*Oxoferryl intermediates’ formation* was monitored using stopped-flow techniques. CcO in the basic medium, pH 8.1 supplemented with 0.1% DM, 0.05 mM ferricyanide and when necessary, T4 was rapidly mixed with an equal volume of the same buffer containing H_2_O_2_.

*Measurement error*. All measurements were repeated 2–4 times. The variation of data between repetitions was about 10%. As a rule, the figures show the results of single typical experiments.

*Data processing* was mainly performed using Origin 7 and 9 Microcal software (https://www.originlab.com/) (accessed on 18 January 2022). For the stopped-flow data processing, Pro Kineticist software provided with the Applied Photophysics SX-20 instrument was also used.

*Kinetic analysis of inhibition data* was performed using the theoretical function
(1)$$v=\frac{1}{1+\frac{\left[I\right]}{K_{i\left(app\right)}}}$$
where *v* is the normalized rate of the reaction, [*I*] is the concentration of the hormone and *K*_i(app)_ is the apparent inhibition constant at a given concentration of DM. 

*Molecular docking* was carried out using the Autodock Vina program.

## 3. Results

### 3.1. Inhibition of Cytochrome Oxidase Activity

#### 3.1.1. Inhibition of Solubilized Enzyme

The effect of thyroid hormones T2, T3 and T4 on the oxidase reaction catalyzed by solubilized CcO from bovine heart mitochondria was studied. We found that hormones T3 and T4 at submillimolar concentrations significantly inhibit CcO activity. The sensitivity to the inhibitor decreased with an increasing concentration of DM, a mild detergent added to keep CcO in a soluble form.

Figure 1 illustrates the main features of the inhibitory effects on CcO activity induced by thyroid hormones.

A typical recording, illustrating deceleration of the rate of oxygen consumption by CcO after addition of T4, is shown in Figure 1A. The reaction proceeded at a constant rate corresponding to the enzyme turnover of about 200 *e*^−^/s. Addition of 0.25 mM T4 to the turning-over enzyme decreased the rate of oxygen consumption by approximately two times. The constant rate reached ~20 s after the addition, which slightly exceeded the mixing time in this experiment (5–10 s). A similar effect on the oxidase activity of solubilized CcO was induced by T3 (data not shown).

The dependence of the normalized CcO activity on the T3 concentration obtained in the presence of 0.02% (0.4 mM) DM is shown in Figure 1B. As seen, the experimental data are well-described by the theoretical hyperbolic function (1) (see Section 2).

We found that similar to steroid hormones [14], the inhibition by thyroid hormones depends significantly on the DM concentration in the medium. The dependence of *K*_i(app)_ values on [DM] is shown in Figure 1C. In the range tested (0.4–10 mM DM), the dependence is close to linear, which points to 1:1 competition between T3 and DM to bind with the enzyme. Data linearization allowed us to determine the true inhibition constant for T3, *K*_i_ ≈ 0.22 mM as well as the dissociation constant of the DM-CcO complex, *K*_c_ ≈ 1.12 mM (see the legend). It should be noted that at high DM concentrations, the titration curves are slightly shifted along the *X*-axis to the right, without changing the values of *K*_i(app)_. A similar effect was observed earlier on steroids [14] and Triton TX-100 [15]. Though the exact origin of this “lag-phase” remains unclear, we assume it to be associated with interaction of the inhibitor and the empty DM micelles.

Figure 1D demonstrates the dependence of normalized CcO activity on the concentration of T4 at [DM] = 0.05% (1 mM). As previously noted, the data are well-approximated by a hyperbolic function that allows us to estimate *K*_i(app)_. It is noteworthy that at the lowest tested concentrations of DM (0.02% and 0.05%), T4 inhibits only 70–80% of enzyme activity, while T3 induces complete inhibition of CcO activity at all tested DM concentrations including 0.02% (*cf*. to Figure 1B). Such an effect of DM concentration on the level of inhibition was observed earlier with testosterone and discussed in [14]. Figure 1E represents the dependence of *K*_i(app)_ values for T4 on [DM]. As with T3 (*cf*. Figure 1C), it is close to linear and enables us to estimate the values of the constants: *K*_i_ ≈ 0.1 mM T4 and *K*_c_ ≈ 1.1 mM DM.

Figure 1F illustrates the effect of T2. In contrast to T3 and T4, it does not actually inhibit the oxidase reaction. A slight suppression of activity is observed only in the millimolar concentration range. The *K*_i_ value for T2 obtained after approximation of the experimental data with the theoretical function (1) is 5.2 ± 0.5 mM. It is symptomatic that unlike T3 and T4 hormones, the inhibitory effect of T2 is not sensitive to DM concentrations in the medium (compare filled and empty symbols corresponding to data obtained at 0.05% and 1% DM). The same results were obtained when DM was replaced with 1% Tween 20 (data not shown). Moreover, we found that *K*_i(app)_ for T2 decreased five times when CcO activity was assayed by ferrocytochrome *c* oxidation, an alternative method that requires a 40–50-times lower enzyme concentration compared to oxygen consumption measurements. A series of control experiments confirmed that changes in the *K*_i(app)_ value were associated precisely with variation of the enzyme concentration rather than other distinctive features of the alternative method (the absence of ascorbate and TMPD or the presence of a small admixture of oxidized cytochrome *c*). The dependence of the effect on the enzyme concentration indicates probable non-specificity of CcO interaction with T2. It is important to note that a similar change in CcO concentration does not induce any changes in the inhibition parameters of hormones T3 and T4.

#### 3.1.2. Inhibition of CcO in the Membrane

First, the effect of thyroid hormones on CcO incorporated into azolectin vesicles was studied. The activity was assayed in the same way as in solution except for the absence of detergent and presence of protonophore in the medium (see Section 2).

The effect of hormones T3, T4 and T2 on the oxygen consumed by proteoliposomes is shown in Figure 2A–C, respectively. Inhibition is observed with all hormones; however, it affects only 50–60% of the activity. The latter is associated presumably with the heterogeneity of the preparation, and in particular, with the presence of multilayer liposomes. The enzyme trapped in the inner layers can oxidize TMPD while remaining inaccessible for inhibition within the period of the activity registration. The data obtained for the titration of the inhibitor-sensitive part of the activity fit well with hyperbolic function (1) and give the *K*_i_ parameters: 0.19 mM T3 (Figure 2A), 0.31 mM T4 (Figure 2B) and 1.4 mM T2 (Figure 2C), which are sufficiently close to the values obtained for the enzyme in solution (see above).

Figure 2D illustrates the oxidase activity of proteoliposomes registered by an alternative method, i.e., monitoring oxidation of the preliminary reduced cytochrome *c* in the absence of ascorbate and TMPD. Pre-incubation of proteoliposomes with 0.5 mM T3 for 30 min results in an approximately twofold decrease in the activity compared to the control, in full agreement with the data obtained on Oxygraph (see Figure 2B). A sufficient time-resolution of the spectrophotometric measurements allows us to note that the starting reaction rates are completely identical in the control (curve *1*) and after pre-incubation of the sample with T3 (curve *2*). In the latter case, inhibition develops only after 10 s of enzyme activity (~450 turnovers) in the presence of the hormone. It can be concluded that (i) the absence of ascorbate and TMPD does not affect inhibition, and (ii) only turning over the enzyme is susceptible to inhibition.

Then, CcO in the native mitochondrial membrane was investigated. As shown in Figure 2E, respiration of rat liver mitochondria was monitored in the presence of T3 at various concentrations. Mitochondria were preliminarily exhausted in endogenous substrates and their outer membrane was partially destroyed, which made it possible to use exogenous cytochrome *c* as an electron donor. The cytochrome oxidase activity was found to be completely sensitive to T3 inhibition. The data obtained are well-described by the hyperbolic function (1). The T3-induced inhibition of CcO in mitochondria is characterized by *K*_i_ value 2.1 ± 0.14 mM, which is 10 times higher than the values obtained for isolated enzymes (0.22 mM, Figure 1B,C) or proteoliposomes (0.19 mM, Figure 2A).

This difference is apparently not related to the tissue-specific properties of the enzyme since similar results were obtained for the inhibition of bovine heart SMP respiration by T4 (Figure 2F). As with mitochondria, complete inhibition was observed. The data well fit function (1) with *K*_i_ = 2.2 ± 0.2 mM, which is an order of magnitude higher than the *K*_i_ values for the enzyme in solution (0.1 mM, Figure 1D,E) or in proteoliposomes (0.31 mM, Figure 2B).

### 3.2. Effect on the Activity of CcO in Eu-Oxidase Phase of Catalytic Cycle: Inhibition of Electron Transfer from Heme a to Heme a_3_

To determine which stages of intramolecular electron transfer are sensitive to thyroid hormones, the effect of T3 and T4 on the level and kinetics of heme reduction was studied (Figure 3). The oxidase reaction was carried out using ascorbate (as an electron source) in combination with the redox mediators RuAm and TMPD. The reduction of CcO was monitored spectrophotometrically.

Panel (**A**) presents the kinetics of absorption growth at 605 nm relative to 630 nm (α-absorption band with a contribution of heme *a* of about 80% [28]). The injection of the respiratory substrate (shown by an arrow) results in an instant increase of absorption up to a steady-state level of heme *a* reduction in the turning-over CcO. After 2–3 min, the reduction sharply rises to 100% due to the depletion of oxygen. It can be seen that in the presence of T3 (1 mM, red curve *2*, or 2 mM, blue curve *3*) or T4 (1 mM, green curve *4*) anaerobiosis occurs later than in the control (black curve *1*). This is explained by the inhibitor slowing down the enzyme activity. Notably, the inhibitory effect of thyroid hormones on CcO does not require the presence of cytochrome *c* and is also manifested in the oxidation of reduced TMPD or RuAm. In addition, the steady-state level of heme *a* reduction in the presence of thyroid hormones is substantially higher than in the control: by ~75% and ~115% in 1 mM and 2 mM T3 (curves *2*, red and *3*, blue), respectively, and by ~50% in 1 mM T4 (green curve *4*).

Panel (**B**) shows the kinetics of the cumulative reduction of hemes *a* and *a*_3_ recorded as the absorption growth at 444 nm relative to 500 nm, i.e., at the γ-bands where the contributions of heme *a* and *a*_3_ to the absorption are approximately equal [28]. It can be noted that enzyme reduction on the onset of anaerobiosis is clearly biphasic, which is consistent with the literature data. The rapid reduction of heme *a* is followed by a much slower phase, which is presumed to be associated mainly with the reduction of heme *a_3_* [29,30,31]. As can be seen, the second phase corresponding to electron transfer to heme *a_3_* noticeably slows down in the presence of 0.5 mM T4 (red curve *2*) compared to the control (black curve *1*). The values of t_1/2_ can be roughly estimated as 5.8 s and 0.9 s, respectively.

### 3.3. Effects of Thyroid Hormones on the CcO Partial Activities Associated with Peroxidase Phase of Catalytic Cycle

In the presence of excess hydrogen peroxide, CcO operates within a truncated cycle, bypassing the eu-oxidase phase [32]. The binuclear center undergoes the transformations: Ox→F_I_-607→F_II_-580→Ox (where Ox designates the free oxidized state and F_I_-607 and F_II_-580 are oxoferryl intermediates similar to those found during oxygen reduction, see reviews [1,2,7,33]), described as a “pseudocatalase cycle” [34]. The reactions of formation and transformation of oxoferryl intermediates make the basis for the partial redox activities catalyzed by CcO: peroxidase, catalase and generation of superoxide.

#### 3.3.1. Thyroid Hormones Do Not Inhibit CcO Peroxidase Activity

Aerobic registration of the peroxidase activity of CcO requires high-potential (E_0_ ≥ +400 mV) electron donors like *o*-dianisidine or ferrocyanide/ferricyanide pair [32,35,36,37]. We found *o*-dianisidine to be inapplicable in the present investigation because its interaction with CcO was severely impaired in the presence of thyroid hormones (data not shown). Therefore, a ferro-ferricyanide redox pair with a potential of ~+420 mV was used. The reaction was followed by formation of ferricyanide. A typical experiment is shown in Figure 4.

As can be seen, addition of 1 mM T4 to the medium before the start of the reaction (red curve *2*) as well as in its course (blue curve *3*) does not noticeably change the rate of ferrocyanide peroxidation compared to the control (black curve *1*). Similar results were obtained with T3 and T2 hormones. The respective rates of peroxidase reaction were: 0.19 *e*^−^/s (control), 0.21 *e*^−^/s (in the presence of 0.5 mM T2), 0.24 *e*^−^/s (in the presence of 0.5 mM T3) and 0.22 *e*^−^/s (in the presence of 0.5 mM T4), with the measurement error about 10%.

#### 3.3.2. Thyroid Hormones Do Not Inhibit Catalase Activity of CcO

Figure 5 shows the effect of thyroid hormones on the catalase activity of CcO.

The decay of peroxide was recorded by the oxygen release. The reaction was triggered by addition of CcO, after which the O_2_ concentration in the medium began to increase at a constant rate. No addition of a thyroid hormone (0.5 mM T2, T3 or T4) to the assay medium preliminary, or in the course of the reaction, significantly affected the rate of O_2_ release (see illustrative curves *1*–*3*). Higher hormone concentrations (curve *4*) caused a slight suppression of the activity, not exceeding 10–20%.

#### 3.3.3. Thyroid Hormones Slightly Accelerate Formation of Spectral Form-580

The formation of intermediates F_I_-607 and F_II_-580 was conveniently monitored spectrophotometrically during titration of the oxidized CcO with increasing concentrations of peroxide [38,39,40,41]. We carried out such a titration to evaluate the effects of thyroid hormones on the equilibrium of oxoferryl intermediates. In the presence of 0.5 mM T3 or T4, a small (within 15–25%) but reproducible decrease in the quasi-stationary concentration ratio [F_I_-607]/[F_II_-580] was observed compared to the control (data not shown).

Qualitatively, the same result was obtained on rapid mixing of oxidized CcO with hydrogen peroxide in a stopped-flow device. Figure 6 shows the results obtained after mixing H_2_O_2_ with CcO in the presence or absence of T4.

Panels (**A**) (control) and (**B**) (experiment in the presence of T4) show kinetic traces of F_I_-607 and F_II_-580 formation. In the presence of T4, F_I_-607 forms slightly slower (t_1/2_ ≈ 0.99 s vs. 0.91 s) and its yield is lower than in the control (*cf.* black curve *1* in panels (**A**, **B**)), whereas the formation of F_II_-580 is accelerated by 23% (t_1/2_ ≈ 0.85 s vs. 1.1 s) and the yield is higher than in the control (*cf.* red curve *2* in panels (**A**, **B**)). Figure 6C shows the difference absorption spectra of the control sample (spectrum *1*) and the sample treated with T4 (spectrum *2*) obtained 15 s after mixing, when the observed difference looks most valid. As seen, in the presence of T4, the concentration of F_I_-607 is lower and F_II_-580 higher than in the control, though the deviation is only about 10%. Accordingly, the overall decrease of the ratio [F_I_-607]/[F_II_-580] is about 20%, exactly as in the titration experiments (see above).

### 3.4. Thyroid Hormones Strongly Inhibit the Generation of Superoxide Anions by CcO

Pseudocatalase cycle turnover is shown to be accompanied by superoxide release [27,42]. The kinetics of superoxide production by CcO, registered by the reduction of WST-1, are shown in Figure 7A.

The instantaneous increase in absorption immediately after the peroxide addition is associated with the transition of CcO to the oxoferryl state. Then, the reduction of the dye proceeds at a constant rate, which drops threefold after addition of 0.4 μM T4 (curve *1*, black). In contrast to the mitochondrial enzyme, CcO from *R. sphaeroides* does not generate O_2_^●−^ under the same conditions (curve *2*, red).

To probe whether the suppression of WST reduction in Figure 7A is due to the superoxide dismutase activity of T4, we tested the effect of T4 on superoxide generation by xanthine oxidase oxidizing hypoxanthine (Figure 7B). The conditions were adjusted so that the rate of WST reduction was approximately the same as in panel (**A**). As seen in Figure 7B, the addition of 500 μM T4 does not affect the rate of dye reduction; instead, as expected, the process is completely blocked by superoxide dismutase.

Similar results are obtained with hormones T2 and T3. The concentration dependences of the inhibition of O_2_^●−^ generation by T2, T3 and T4 are shown in panels (**C**), (**D**) and (**E**), respectively. All three hormones induced a complete inhibition of the reaction, and the experimental data fit well with function (1). The *K*_i_ values obtained by approximation are: 2.6 μM T2, 5.1 μM T3 and 0.3 μM T4.

## 4. Discussion

As we have found recently, a number of steroid-like compounds have a pronounced inhibitory effect on solubilized CcO from mitochondria [14,15]. The inhibition is characterized by a 10^−5^–10^−4^ M *K*_i_ range and 1:1 stoichiometry competition between the inhibitor and DM. We assumed that steroid-like inhibitors may be ligands of the BABS site and their inhibitory effect associated with impaired conductivity of the proton channel K. Our assumption was confirmed by the fact that steroid binding arrests the stages of the catalytic cycle at which channel K is active (eu-oxidase phase), while the stages at which it is closed (peroxidase phase) remain unaffected. We also assumed that DM in our experiments would substitute an endogenous phospholipid molecule that is tightly bound near BABS and partially overlaps with it in some conformations [9,11].

The data presented in this work indicate that thyroid hormones are able, like steroids, to interact with mitochondrial CcO at BABS and modulate the enzyme activity. Indeed, thyroid hormones T3 and T4 effectively inhibit the oxidase activity of a solubilized enzyme with a *K*_i_ of about 200 μM and 100 μM, respectively (Figure 1C,E), and the apparent *K*_i_ values increase with an increase in the DM concentration. Inhibition data analysis points to the competition between DM and the inhibitor in a 1:1 ratio, and noteworthy, predicts the dissociation constant for the DM-CcO complex to be about 1.1 mM (see the *K*_c_ value in Figure 1C,E). The latter is rather close to the values obtained earlier in the studies on CcO inhibition by estradiol and testosterone (*K*_c_ = 1.47 mM and 1.3 mM, respectively) [14] and by Triton X-100 (*K*_c_ = 1.2 mM) [15]. This coincidence is a serious indication that thyroid and steroid hormones bind to CcO at the same place, namely, in the region where the enzyme affinity for DM is characterized by a dissociation constant of 1–1.5 mM.

Like steroids, thyroid hormones T3 and T4 do not inhibit the partial CcO activities associated with the peroxidase phase of the enzyme cycle. At concentrations sufficient for almost complete suppression of the oxidase activity, hormones T3 and T4 do not reduce the peroxidase activity of CcO (see Figure 4, which demonstrates the complete resistance of peroxidase activity to 1 mM T4, that is, 5 *K*_i(app)_ under the conditions of the experiment, see Figure 1E). The catalase activity of CcO is also actually insensitive to the action of T3 and T4 at concentrations of 0.5–1 mM (Figure 5).

The generation of superoxide is another partial reaction associated with the peroxidase phase of the CcO catalytic cycle, and is worth a separate mention. This activity is not affected at all by sex steroid hormones (unpublished data), while thyroid hormones, unexpectedly, inhibit it with high specificity (Figure 7). The reaction conditions, however, suggest that in the latter case, O_2_^●−^ is formed on the enzyme surface rather than in the binuclear center (see below for the details).

At the same time, a small but reliable effect of 0.5 mM T3 and T4 on the interconversion of intermediates F_I_-607 and F_II_-580 formed in the reaction of CcO with peroxide has been registered both by steady-state titrations and the rapid-mixing technique (Figure 6). This manifests mainly in a 20–25% accelerated formation of an intermediate with spectral properties of form 580 (compare red curve *2* in Figure 6A,B). As seen in Figure 6B, during the first second of the rapid-mixing experiment, the kinetics of the appearance of two oxoferryl forms in the presence of 0.5 mM T4 coincide. This allows us to propose that intermediates with absorption maxima at 607 nm and 580 nm are not formed in this case in the Ox → form 607 → form 580 sequence as in the control (Figure 6A), but rather synchronously from Ox. Therefore, we hypothesize that in the presence of T4, at least part of the observed “form 580” actually represents intermediate-I isoelectronic to F_I_-607 but spectrally similar to F_II_-580. Our interpretation is clearly supported by the fact that the yield of spectral “form 607” (black curve *1*) in the presence of T4 looks noticeably lower and that of spectral “form 580” (red curve *2*) higher than in the control. The specific kind of intermediate-I with a broad absorption band of around 570–580 nm is known as the “acidic” form, or F′ [43], or F^●^ [44], or F-I_575_ [45]. It is observed during the reaction of CcO with peroxide at an acidic pH and is believed to differ spectrally from F_I_-607 due to protonation of a certain group near the binuclear center [44]. It is of interest that K362M replacement of the key residue in the K-channel makes the spectral properties of intermediate-I pH-independent [45]. It can be assumed that T3 or T4 binding at BABS changes the structure of hydrogen bonds in the K-channel, which leads to protonation of the group that determines the spectral properties of intermediate-I. Note, however, that the observed effect is not large: the F_I_-607/F_II_-580 ratio in the presence of 0.5 mM T3 or T4 changes by no more than 20% compared to the control (see Figure 6C). Perhaps the described alteration affects only a small fraction of the enzyme.

Thyroid hormones T3 and T4 inhibit the transfer of the first two electrons to the oxidized binuclear center, the process related to the eu-oxidase part of the catalytic cycle. Firstly, this is indirectly evidenced by an increase in the steady-state level of heme *a* reduction while the enzyme is operating in the presence of T3 or T4 (Figure 3A). This can be explained by the slower escape of an electron from heme *a* to the binuclear center. Secondly, hormones T3 and T4 noticeably affect the kinetics of CcO reduction during the onset of anaerobiosis (Figure 3B). The total reduction of the hemes during the transition of the enzyme to a completely reduced state is a two-phase process, with the fast phase corresponding mainly to the reduction of heme *a*, and the slow phase to the reduction of heme *a_3_* [29]. In the presence of T3 or T4, the slow component of hemes’ reduction decelerates noticeably, which indicates inhibition of electron transfer to heme *a_3_*.

Thus, thyroid hormones T3 and T4 exhibit the same properties of BABS ligands and K-channel inhibitors as steroid hormones [14]. Figure 8 shows the structure of CcO in the BABS region with estradiol (panel (**A**)) or triiodo-thyronine (panel (**B**)) harbored by the site. As can be seen, both molecules are located in the hydrophobic cavity at approximately the same level, in the immediate vicinity of the entrance to the K-channel.

In contrast, the thyroid hormone T2, though also inhibiting oxidase activity, differs drastically from T3 and T4 in its effect on the enzyme. Firstly, the effective T2 concentrations are tens of times higher (the formal *K*_i(app)_ value is 5.22 ± 0.5 mM). Secondly, and extremely significant, the sensitivity to T2 does not depend on the DM concentration (compare open and filled symbols in Figure 1F). Third, it turned out that the inhibitory effect of T2 on oxidase activity strongly depends on the enzyme concentration in the assay: a decrease in CcO concentration by 40–50 times causes a decrease in the formal *K*_i(app)_ value from 5.22 mM to 0.89 mM. Judging on the latter feature, the T2-induced effect appears to be non-specific. It should be concluded that the T2 hormone, despite its chemical similarity with T3 and T4, does not bind to BABS. Conspicuously, the presence of at least one iodine substituent in the second ring of the ligand molecule is critically important for the interaction of thyroid hormones with BABS. It is of interest that the pharmacological use of the “classical” thyroid hormones T3 and T4 is limited by their toxic effects (especially on the heart), which are much less pronounced in 3,5-diiodo-thyronine [19]. It can be assumed that, among other reasons, the lower toxicity of T2 might be associated with its inability to modulate the CcO activity as a ligand of BABS.

A question remains about the physiological relevance of our results with regard to effective concentrations. In the case of steroids, we already answered a similar question in the affirmative [14] since the *K*_i_ values obtained by us (tens to hundreds of μM) fall well into the range of expected intramembrane concentrations of steroid hormones when taking into account their plasma concentration (*ca.* 10^−8^ M [46,47,48]) and distribution coefficient between the hydrophobic and water phases (*ca.* 10^4^ [49]). The plasma concentration of free thyroid hormones is three orders of magnitude lower than that of steroid hormones (about 5 pM T3 and 12 pM T4 [50]); however, it should be taken into account that the content of thyroid hormones in tissues is much higher than in the blood. Thus, in the work of Donzelli et al., the following contents of T3 and T4 in different organs of the rat were given (in pmol/g): heart—5.2 and 27.9, liver—12.9 and 124.3, kidneys—19.75 and 90.9 [51], respectively. Taking into account the distribution coefficients between octanol and water (1.7 × 10^3^ for T3 and 2.4 × 10^3^ for T4, as computed by XLogP3 3.0, see https://pubchem.ncbi.nlm.nih.gov/ (accessed on 18 January 2022)) and the density of the heart tissue (1.265 g/mL [52]), this gives the following estimates of the hormone concentrations in the lipid environment for rat heart cells: 11.2 μM T3 and 84.7 μM T4. In the rat liver and kidneys, the concentrations of T3 and T4 are 2–4 times higher (see above), and in hyperthyroidism and similar pathologies, the concentrations of both hormones can increase several times more. Thus, the *K*_i_ values obtained in the present work for the enzyme in solution and proteoliposomes approach the possible physiological concentrations of T3, and especially T4, in the inner mitochondrial membrane. This allows us to speculate that the interaction of thyroxine (and probably triiodo-thyronine, as well) with BABS might actually be related to the regulation of CcO in vivo.

In our results obtained on proteoliposomes, one can notice an inconsistency with the above hypothesis. Assuming that the hormone concentration within the liposome membrane is three orders of magnitude higher than in solution (in accordance with the distribution coefficient, see above), one would expect a significant shift of *K*_i(app)_ on proteoliposomes toward lower values compared to the solubilized enzyme. However, the values obtained for CcO in solution and the liposome membrane are similar (0.22 mM vs. 0.19 mM for T3 and 0.1 mM vs. 0.31 mM for T4, see Figure 1B,D, and Figure 2A,B). A possible explanation for this contradiction may be associated with a change in the quaternary structure of the enzyme. Figure 8 shows the structure of BABS in the dimer of CcO; it can be seen that both monomers participate in the arrangement of the site. The solubilized CcO preparation in our experiments contained a small admixture of cholate (1–2 mM), which was used in isolation (Section 2). According to Musatov et al. [53], under these conditions, the enzyme is almost completely represented just by the dimeric form. During the preparation of proteoliposomes, cholate is completely removed (see Section 2), which may result in dissociation of the enzyme into monomers. Obviously, BABS of the monomeric form still binds ligands (as the example of the bacterial enzyme shows [9]); however, the affinity for them can undoubtedly change. This effect may cause the absence of the expected decrease in the effective concentrations of thyroid hormones on proteoliposomes. If this assumption is acceptable, we can try to look at the hormones T3 and T4 as potential players in the complex system of multifactorial regulation of CcO currently suggested [54]. In any case, study of the BABS ligands’ interaction with CcO in proteoliposomes needs to be continued.

Another crucial question that remains obscure is the nature of the main BABS ligand in vivo. We are currently considering two possibilities. First, the “true” BABS ligand, physiological regulator of CcO activity, could be still unknown. This might be a compound of the mitochondrial membrane lost during the purification of the enzyme. Comparison of the CcO sensitivity to thyroid hormones in solution and mitochondrial membrane suggests that the native environment of the enzyme makes it significantly less sensitive to the inhibitor. Thus, the *K*_i(app)_ value for T3 is an order of magnitude higher in the whole mitochondria (2.12 mM, Figure 2E) than in the solubilized enzyme (0.22 mM, Figure 1C). A modulating effect on the CcO sensitivity to the hormone is evidently exerted by some component of the membrane rather than the mitochondrial matrix since a similar increase in *K*_i(app)_ (2.2 mM, Figure 2F vs. 0.1 mM, Figure 1E) is observed in SMP.

The observed change in the CcO sensitivity to inhibition by T3 and T4 in the mitochondrial membrane could be explained by the competition of the thyroid hormone with an unknown endogenous ligand of BABS. However, another explanation is also possible. Dodecyl-maltoside, which in experiments on the isolated enzyme significantly modulates the affinity of BABS to steroid-like ligands and thyroid hormones T3 and T4, may be an analog of a natural lipid [55]. Indeed, a molecule of tightly bound phospholipid is present in 3D structures of mitochondrial CcO in the immediate vicinity of BABS [9], whereas in the enzyme from *R*. *sphaeroides*, a similar place is occupied by the exogenous decyl-maltoside molecule, the alkyl tail of which lies in the groove between subunits I and II, and the maltoside group is able to partially overlap with the putative binding site of amphipathic ligands [11]. We believe that the competition between hormone inhibitors and DM mimics the natural situation in which a tightly bound endogenous phospholipid controls (by a change in conformation, and as a result, the area of overlap with BABS) the effective affinity of BABS to physiologically active amphipathic ligands. The conformation of such a regulatory molecule can, in turn, depend on numerous factors, in particular, on the immediate molecular environment of the enzyme. It is possible that under the conditions of our experiments on mitochondria and SMP, the tightly bound phospholipid interacts quite strongly with BABS, preventing hormone binding. As a result, the affinity of the hormones for the enzyme decreases by an order of magnitude compared to the situation in solution. We do not exclude that in vivo the lipid molecule could assume a different conformation, which results in much more efficient binding of the regulating hormones to BABS. In light of these considerations, the second, and in our opinion, more exciting hypothesis about the native BABS ligand is that the site is capable of interacting with a fairly wide range of amphipathic compounds many of which—including steroid hormones, and probably, thyroid hormones, as well—may be involved in the regulation of oxidase activity.

Signaling pathways that determine the action of thyroid and steroid hormones in the cell intersect [16,17,18,19]. Interaction with CcO at BABS could represent the physical point of such an intersection. We believe that the multispecificity of BABS, which apparently can also be referred to bacterial CcO, may indicate the ancient origin of the regulatory mechanism based on the direct interaction between the key enzyme of oxidative phosphorylation and physiologically active agents, which subsequently, in the course of evolution, became regulators of a high hierarchy.

Quite unexpectedly, it turned out that thyroid hormones inhibit with high specificity the generation of superoxide catalyzed by mitochondrial CcO in the presence of excess peroxide. This phenomenon is determined not by dismutase activity, as one might suspect, since thyroid hormones do not reduce the level of superoxide released during the oxidation of hypoxanthine by xanthinoxidase (Figure 7B). Inhibition of superoxide formation by thyroid hormones is fundamentally different from the suppression of oxidase activity by hormone ligands of BABS in two aspects: (i) inhibition is induced not only by T3 and T4 but also and to no lesser extent by the T2 hormone; (ii) the effective concentrations of hormones are two to three orders of magnitude lower (*K*_i_ values are 2.6 μM for T2, 5.1 μM for T3 and 0.3 μM for T4—see Figure 7C–E). These differences allow us to conclude that the inhibitory effect of T2, T3, and T4 on superoxide generation is associated not with BABS but with some other site of interaction characterized by a much higher affinity for thyroid hormones. In this regard, it should be recalled that O_2_^●−^ molecules formed in the reaction of oxidized CcO with peroxide may originate from more than one source. Firstly, they are released by the binuclear center at the stages of the pseudocatalase cycle that correspond to the transformations F_I_-607→F_II_-580 and F_II_-580→Ox (see Scheme 1 in [34]). Secondly, according to [56], the lipid-based (or probably amino acid-based) radicals occur at the surface of CcO on its interaction with peroxide, and these groups become another site of peroxide decomposition. It is assumed that such free radicals appear after a certain number of pseudocatalase cycle turnovers as a result of the migration of an electron vacancy, which appeared initially near the binuclear center, to the protein surface. It is easy to suggest these radical(s) react with molecular oxygen, forming superoxide. Therefore, we assume that the second place that binds thyroid hormones with high affinity and is responsible for suppression of superoxide formation is localized on one of the outer small subunits of the mitochondrial enzyme. It is noteworthy that bacterial CcO lacking additional subunits does not form O_2_^●−^ under the same conditions (red curve *2* in Figure 7A). The mechanism of inhibition could consist, for example, of shielding a free radical group on the surface of the enzyme by a hormone molecule, thus preventing contact with O_2_. Quite probably, this scenario should not be regarded as associated with the pseudocatalase cycle exclusively since the chance of an electron vacancy migration to the protein surface exists even in CcO operating along a complete catalytic cycle.

It is of interest that the first data on the direct interaction of hormones with mitochondrial CcO were obtained by the groups of Goglia and Kadenbach in the mid-1990s and were related just to the thyroid hormone T2 [57,58]. According to the authors, T2 binds with extremely high (micromolar-submicromolar) affinity to the Va subunit of mitochondrial CcO. This causes small changes in the absorption spectrum of the enzyme and its activation by about one and a half times as a result of the release from allosteric inhibition exerted by ATP molecules interacting with the matrix domain of the subunit IV. We were unable to reproduce the indicated effects of T2, most likely due to differences in the methods of enzyme isolation used in [57] and in our study. Nevertheless, we assume that in the case of inhibition of superoxide formation by thyroid hormones, the same site on the Va subunit may be concerned. As a result of interaction with it, we observed the inhibition of one of the partial activities of the enzyme rather than modulation of its main activity described by the Goglia and Kadenbach groups, and the inhibiting agent in our case was not only T2 but T3 and T4, as well.

The extremely high affinity of thyroid hormones for the “second” site is remarkable. Goglia et al. [57] reported on the effects caused by T2 starting with a concentration of 10^−7^ M at 0.2 μM enzyme in the experiment. As shown in Figure 7E, T4 also produces an effect in the submicromolar concentration range (*K*_i_ = 0.3 μM), with an enzyme concentration of 0.5 μM. Judging from these data, the complex of thyroid hormone with CcO is very stable, and the induced inhibition of activity (superoxide formation) is close to stoichiometric. Recall that according to our calculations, T3 and T4 can be present in the inner membrane of cardiac mitochondria at concentrations of the order of 10 and 100 μM, respectively (see above). This means that under certain physiological conditions, the thyroid hormone molecule (most likely, T4) will be in complex with CcO constantly, actually as an enzyme cofactor. Although the question of the physiological role of the second binding site for thyroid hormones in mitochondrial CcO remains open, the high affinity hints that such a role exists. It could just be protective since superoxide, and especially its derivatives, damage intracellular structures chemically, with mitochondrion to be the most sensitive target (see, for example, [59]). Furthermore, O_2_^●−^ is known as a link of intracellular signaling pathways [60,61], and the ability to prevent its formation makes thyroid hormones potential participants in the redox regulation system.

## Figures and Tables

**Figure 1 cells-11-00908-f001:**
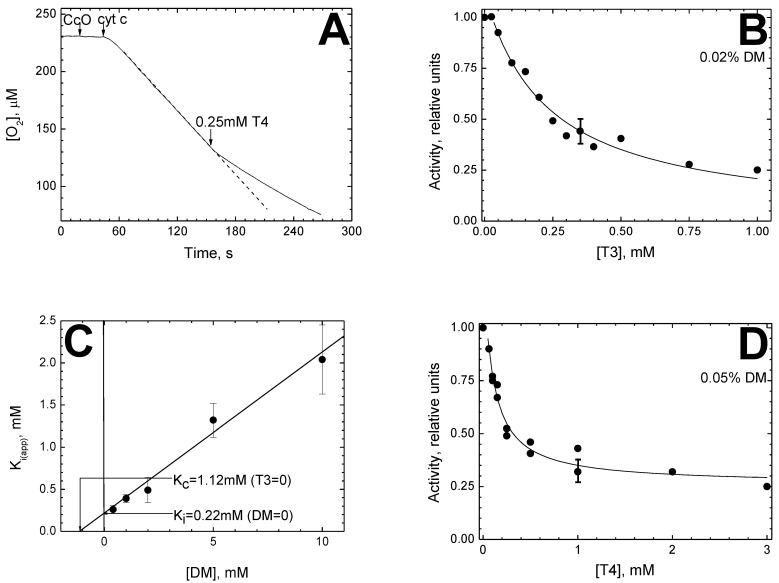
Inhibition of the solubilized cytochrome *c* oxidase (CcO) activity by thyroid hormones. (**A**) Inhibition of oxygen consumption. Additions of CcO (20 nM), cytochrome *c* and T4 (0.25 mM) are shown by the arrows. The kinetics trace is corrected for ascorbate autooxidation. To highlight the inhibitory effect, the trace in the absence of T4 is shown by the dotted line. (**B**) Titration of CcO activity by T3 at 0.02% dodecyl-maltoside (DM). Other conditions are mainly as in panel (**A**). Experimental data (circles) are approximated by function (1), see Section 2. The activity (the ratio of the reaction rate after the onset of inhibition to the initial reaction rate) is given in relative units. To streamline the figure, a typical measurement error is shown for only one of the experimental points (hereinafter, in similar cases). (**C**) Dependence of an apparent *K*_i_ for T3 on the DM concentration. The *K*_i(app)_ values are determined from the approximation of the experimental data by function (1), see panel (**B**). The segment being cut off on the *Y*-axis indicates the true inhibition constant *K*_i_ for T3 in the absence of DM; the segment being cut off on the *X*-axis in its negative area indicates the value of the dissociation constant for DM in the absence of T3, *K*_c_ (both segments are pointed out by arrows). (**D**) Titration of CcO activity by T4 at 0.05% DM. See panel (**B**) for further details. (**E**) Dependence of an apparent *K*_i_ for T4 on the DM concentration. The *K*_i(app)_ values are determined as described above, see panel (**C**). (**F**) Titration of CcO activity by T2 at 0.05% (filled signs) and 1% (open signs) DM. Other conditions are mainly as in panel (**A**). The entire dataset is approximated by function (1).

**Figure 2 cells-11-00908-f002:**
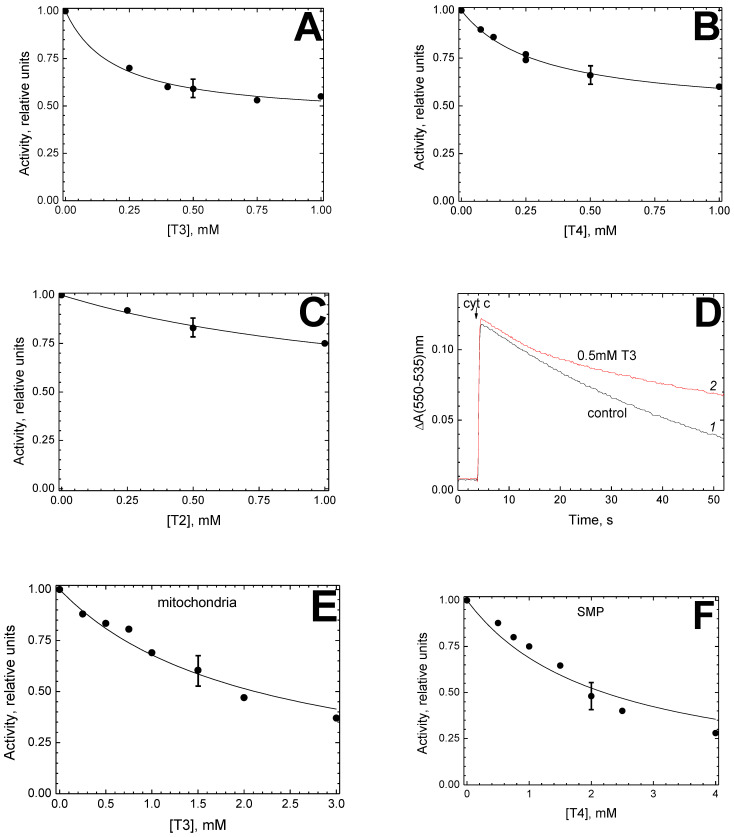
Inhibitory effect of thyroid hormones on the oxidase activity of the membrane-incorporated CcO. (**A**–**C**) Titration of the oxidase activity of CcO in proteoliposomes by T3 (**A**), T4 (**B**) and T2 (**C**). Proteoliposomes were added to the experimental medium up to 26 nM CcO. The oxidase reaction was initiated by addition of the respiratory substrate. Other details are as in Figure 1B,D. (**D**) T3-induced inhibition of ferrocytochrome *c* oxidation by CcO in proteoliposomes. The activity was measured in the absence (control curve *1*, black) or in the presence (curve *2*, red) of 0.5 mM T3 in the medium. Other conditions are as in panel (**A**) except that proteoliposomes were added to the medium up to 3.25 nM CcO and pre-incubated for 30 min before adding the respiratory substrate. (**E**) Inhibitory effect of T3 on CcO oxidase activity in rat liver mitochondria. Mitochondria were suspended in the experimental medium up to 0.8 mg of protein/mL. Other details are as in Figure 1B,D. (**F**) Inhibitory effect of T4 on CcO oxidase activity in SMP obtained from bovine heart mitochondria. The conditions were essentially as in panel (**E**) except that the CcO concentration was 20 nM.

**Figure 3 cells-11-00908-f003:**
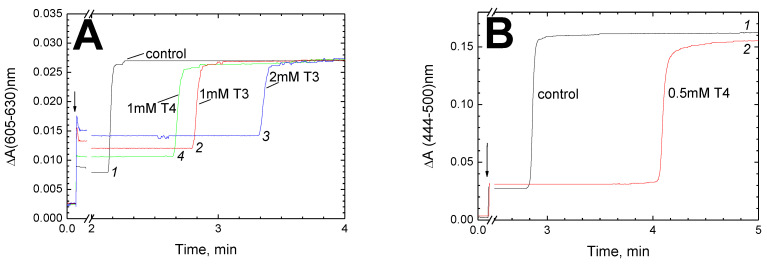
Hormones T3 and T4 decelerate electron transfer from heme *a* to heme *a_3_*. (**A**) Effect of T3 and T4 on the steady-state level of heme *a* reduction. CcO (*ca*. 1 μM) in the basic medium pH 8.0 supplied with 5 μM RuAm, and where indicated, T3 or T4 was placed in a closed cuvette and reduced by addition of 5 mM ascorbate (indicated by the arrow). The reduction level of heme *a* was followed in the absence (control trace *1*, black) or presence of 1 mM T3 (trace *2*, red), 2 mM T3 (trace *3*, blue) or 1 mM T4 (trace *4*, green). (**B**) Effect of T4 on the kinetics of the reduction of hemes *a* and *a_3_* on the onset of anaerobiosis. The total reduction level of hemes *a* and *a_3_* was registered. Trace *1*, black = control and trace *2*, red = reduction of the sample in the presence of 0.5 mM T4. The other conditions were as in panel (**A**) except that 0.1 mM TMPD was used instead of RuAm.

**Figure 4 cells-11-00908-f004:**
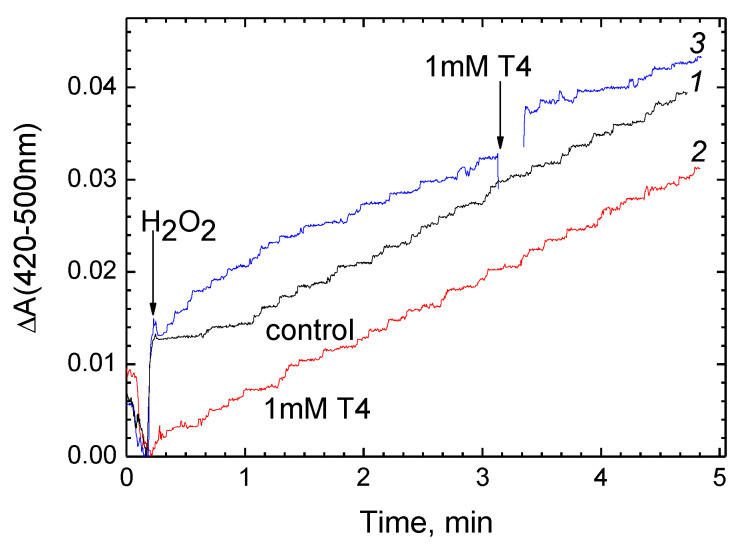
Peroxidase reaction catalyzed by CcO is not affected by thyroid hormones. Peroxidation of ferrocyanide (0.2 mM) was monitored in the presence of 0.6 μM CcO. The reaction was triggered by addition of 4 mM H_2_O_2_ (shown by the arrow). Trace *1*, black = control; trace *2*, red = the experiment in the presence of 1 mM T4; trace *3*, blue = 1 mM T4 added in the course of the experiment, as indicated. The initial jump on H_2_O_2_ addition reflects spectral response in the γ-band of heme *a_3_* on oxoferryl intermediates’ formation.

**Figure 5 cells-11-00908-f005:**
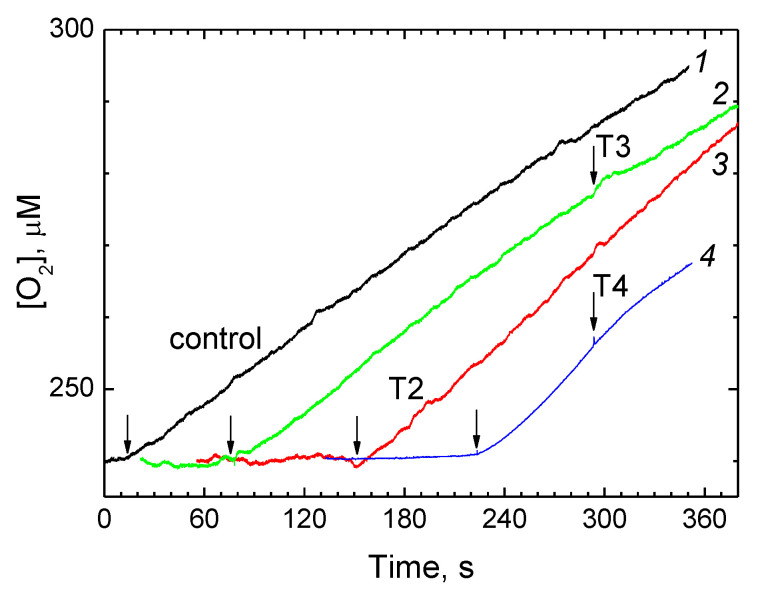
Thyroid hormones do not inhibit catalase activity of CcO. Pre-addition of 12 mM H_2_O_2_ to the experimental medium did not induce detectable oxygen release. Additions of 1.5 μM CcO are indicated by the vertical arrows. Trace *1*, black = control; traces *2*–*4* = thyroid hormones (T3 = *2*, green; T2 = *3*, red; T4 = *4*, blue). T2 was pre-added up to 0.5 mM; T3 and T4 were added up to 0.5 mM and 1 mM, respectively, as marked.

**Figure 6 cells-11-00908-f006:**
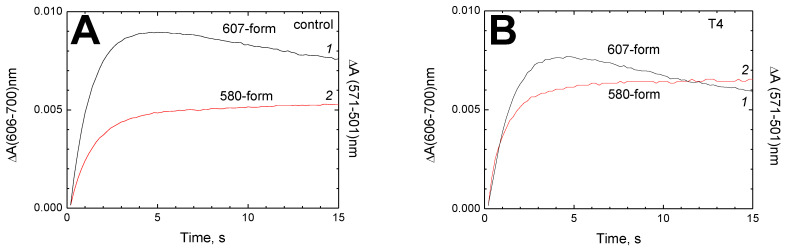
Effect of T4 on the formation of CcO oxoferryl intermediates during the reaction with H_2_O_2_. The reaction was followed using a diode array spectrophotometer. Final concentrations after mixing: 3.25 μM CcO, 1 mM H_2_O_2_, 0.5 mM T4. (**A**, **B**) Kinetics of the oxoferryl intermediates’ formation. Spectral forms F_I_-607 (trace *1*, black) and F_II_-580 (trace *2*, red) are followed by the absorption differences at 606–700 nm and 571–501 nm, respectively. (**A**) Control; (**B**) CcO was pre-incubated with 0.5 mM T4 until mixing. (**C**) Difference spectra (15 s vs. 0.2 s after mixing) obtained in the absence (control spectrum *1*, black) and presence (spectrum *2*, red) of 0.5 mM T4.

**Figure 7 cells-11-00908-f007:**
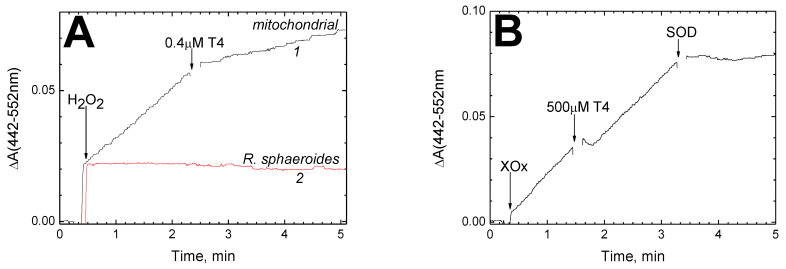
Thyroid hormones strongly inhibit superoxide generation by CcO in the presence of excess H_2_O_2_. (**A**) Typical kinetics of superoxide production catalyzed by mitochondrial CcO (trace *1,* black) as compared to bacterial *aa*_3_-type oxidase from *R. sphaeroides* (trace *2*, red). Addition of 0.4 µM T4 is indicated. CcO was added up to 0.5 μM to the experimental medium, supplemented with 0.1 mM WST-1. The reaction was initiated by addition of 4 mM H_2_O_2_ (indicated by the arrow). (**B**) T4 has no superoxide dismutase activity. Superoxide formation accompanying the oxidation of hypoxanthine (50 μM) by xanthine oxidase (XOx, 0.015 units/mL) was followed as in panel (**A**). Additions of 500 μM T4 and superoxide dismutase (SOD, 20 μg/mL) are shown by the arrows. (**C**–**E**) Concentration dependence of the inhibitory effect of thyroid hormones T2 (**C**), T3 (**D**) and T4 (**E**), respectively, on superoxide production by mitochondrial CcO. Conditions of the measurements are as in panel (**A**). Experimental data are approximated by function (1) with different values of the *K*_i_ parameter (indicated).

**Figure 8 cells-11-00908-f008:**
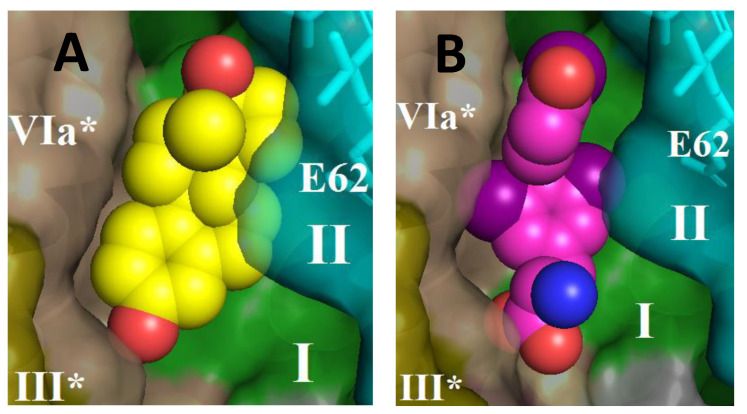
Proposed interaction of estradiol (**A**) and triiodo-thyronine (**B**) molecules with CcO in BABS. The structure of the dimeric enzyme in the region of BABS is shown. Side view: the inner (matrix) surface of the membrane is at the bottom. The ligand molecule is docked in a hydrophobic cavity near the entrance to the proton channel K. Subunits I (green) and II (cyan) are shown, as well as subunits III (brown) and VIa (wheat) from the neighboring monomer (marked with an asterisk). The indicated residue E62 from subunit II is located just on the border of the matrix and the entrance to the K-channel.

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
