# Peer review of "Direct Interaction of Mitochondrial Cytochrome c Oxidase with Thyroid Hormones: Evidence for Two Binding Sites"

_cells, 2022, doi:10.3390/cells11050908_

Round 1

Reviewer 1 Report

It is an interesting paper corresponding to the former article (Oleynikov IP, Azarkina NV, Vygodina TV, Konstantinov AA. Interaction of Cytochrome C Oxidase with Steroid Hormones. Cells. 2020; 9(10): 2211) and not at least a continuation of the works  of the Goglia/Kadenbach group. They found Diiodothyronines stimulate the activity of isolated cytochrome c oxidase (CytOx) from bovine heart mitochondria. Maximal stimulation of activity (about 50%) was obtained with 3,3'-T2 at pH 6.4 and with 3,5-T2 at pH 7.4. In contrast, 3,5,3'-triiodothyronine (T3) exhibited no or little stimulation of  the enzymatic activity. Binding of the hormones to CytOx leads to conformational changes as shown by modified visible spectra of the oxidized enzyme. It is suggested that 'short-term' effects of thyroid hormones on mitochondrial respiration are at least partly due to the allosteric interaction of diiodothyronines with the CytOx  complex. (Goglia F, Lanni A, Barth J, Kadenbach B. Interaction of diiodothyronines with isolated cytochrome c oxidase. FEBS Lett. 1994; 346(2-3):295-8)

Susanne demonstrated specific binding of labelled 3,5-diiodothyronine to subunit Va of cytochrome-c oxidase from bovine heart. She found 3,5-Diiodothyronine, and to a small extent triiodothyronine, but not thyroxine and thyronine, abolish the allosteric inhibition of ascorbate respiration of reconstituted cytochrome c oxidase by ATP. This abolition of ATP-inhibition by 3,5-diiodothyronine was completely prevented by a monoclonal antibody to subunit Va. (Arnold S, Goglia F, Kadenbach B. 3,5-Diiodothyronine binds to subunit Va of cytochrome-c oxidase and abolishes the allosteric inhibition of respiration by ATP. Eur J Biochem. 1998 Mar 1;252(2):325-30) But in this case we have ask for the subunit composition of CytOx in the experimental setting.

Why do the authors use bovine soluble CytOx for the first experiment and then CytOx from Rat liver in the next? We have species- dependent molecules! Moreover, enzymatic activities of Heart CytOx and Liver CytOx differ as well!

Sinkler CA, Kalpage H, Shay J, Lee I, Malek MH, Grossman LI, Hüttemann M. Tissue- and Condition-Specific Isoforms of Mammalian Cytochrome c Oxidase Subunits: From Function to Human Disease. Oxid Med Cell Longev. 2017;2017:1534056

Vijayasarathy C, Biunno I, Lenka N, Yang M, Basu A, Hall IP, Avadhani NG. Variations in the subunit content and catalytic activity of the cytochrome c oxidase complex from different tissues and different cardiac compartments. Biochim Biophys Acta. 1998;1371(1):71-82

Arnold S, Lee I, Kim M, Song E, Linder D, Lottspeich F, Kadenbach B. The subunit structure of cytochrome-c oxidase from tuna heart and liver. Eur J Biochem. 1997;248(1):99-103

Yoshikawa used all the time cholate for cristallization  of CytOx, so dialysis was all the time needed. Cholate, however, dimerizes the enzyme isolated in non-ionic detergents and induces a structural change as evident from a spectral change. When 1.3 % Cholate for liposomes in the present paper is used, then it effects the enzymatic activity of CytOx for sure. It is difficult to describe effects of binding sites under these conditions. Please, explain. Moreover, Cholate induces a more or less “dimeric structure” of the enzyme with consequences for its activity. Finally, When the authors postulate Cholate binds to the binding sides for Bile Acids, then we have to think about the fact of competition between Cholate and ADP/ATP. (Ramzan R, Napiwotzki J, Weber P, Kadenbach B, Vogt S. Cholate Disrupts Regulatory Functions of Cytochrome c Oxidase. Cells. 2021; 10(7): 1579) Please, clarify.

Last but not last, all the experiments  in the paper were performed with pH= 7.6  (reconstruction), pH=7.4 (isolation Rat Liver mitochondria), pH= 7.6 (enzymatic activity, superoxide generation). More basic pH- values pronounce the monomeric form of CytOx (Shinzawa-Itoh K, Sugimura T, Misaki T, Tadehara Y, Yamamoto S, Hanada M, Yano N, Nakagawa T, Uene S, Yamada T, Aoyama H, Yamashita E, Tsukihara T, Yoshikawa S, Muramoto K. Monomeric structure of an active form of bovine cytochrome c oxidase. Proc Natl Acad Sci U S A. 2019 Oct 1;116(40):19945-19951). The dimeric form appears at least as the “more physiological enzyme”. So, we would appreciate if the authors could discuss the points and revise the present manuscript.

Reviewer 2 Report

The manuscript by Vygodina and co-workers reports a series of kinetic analyses in which they study the effects of thyroid hormones on the functionality of cytochrome c oxidase. Authors claim that “the mitochondrial enzyme contains two sites of interaction with thyroid hormones”. This conclusion is indeed interesting, yet I lost illusion as I read the manuscript and it made me doubt about the physiological relevance of the data for the first site.

As authors refer from literature, thyroid hormones concentrations in serum are of the order of 1-10 pM, whereas inhibition constants reported in the manuscript with the isolated enzyme are in the range of 100 μM. Then authors argue that data could be conciliated by considering a hydrophobicity/hydropathy partitioning constant of 1e4, which would lead to an effective inhibition constant of 100 nM, still 3 orders of magnitude far from the observed inhibition constant. Then, experiments with proteo-liposomes and, later, isolated mitochondria—in which there is a substantial amount of lipids to favour the hydrophobic interactions—yield even slightly higher values for Ki, that is farther from expected physiological values such as those for nuclear thyroid-hormone receptors, whose dissociation constants are in the pM range. These results do not support the argument of hydrophobic partitioning to attribute a physiological role to the “first” binding site.

Same applies to the intramolecular electron transfer step: Authors show clear results but, again and sadly, at high T4 concentrations. I miss a discussion regarding the mechanism of inhibition. Maybe T4 is somehow acting as an electronic sink—an alternative substrate—instead of an inert molecule.

Inhibition of superoxide generation experiments show that thyroid hormones act at lower concentrations, consistent with authors’ claim of a “second” site, if the first one is considered physiological. Data is consistent with previous reports from other authors indicating in vivo. Contrary to the first CcO activities, authors only display data obtained with isolated proteins, neither with proteo-liposomes nor isolated mitochondria.

As regards binding sites, it would be interesting authors show thyroid hormones dock into the Bile Acid Binding Site and compare it to any steroid—for instance the bile acid structure. Bile acids show puckered hydrophobic rings, whereas thyroid hormones are composed of flat aromatic rings that may interact in a different manner.

Other comments:

Please scale up the graphics and labels, so they become more readable.

Figure 1: Inverse, Lineweaver-Burk representations should be inserted in panels B and F.

Please, indicate how many times experiments were repeated and indicate data errors in the scatter plots.

Reviewer 3 Report

The manuscript by Ilya P. Oleynikov an co-authors describes the direct interaction of mitochondrial Cytochrome c oxidase with thyroid hormones. Overall I found the manuscript to be interesting, however it is very lengthy. Due to that the manuscript seems out of focus. For example the whole 2nd paragraph of the introduction, describing the catalytic cycle of CcO is unnecessary long. Figure 1 description has a lot of the same text for each panel. Although it is long it lacks the description of the observed effects and just focuses on technical details. I suggest to trim the whole text. Also I suggest to re-shift the focus toward the last result. When I was reading the results the major issue for me was the biological relevance. The obtained Ki values, assuming no DM presence, are in sub mM values e.g. 100 uM. The normal blood concentration of T4 for example is 5-12 ug/dL. The upper limit corresponds to 0.15 uM. I could not find a paper describing the level of T4 in cells (though we know that various tissues would have different levels). However I doubt that the cells would accumulate the hormone in 1000x higher concentration then in blood to have an inhibitory effect. My concerns were further were confirmed as the Ki in isolated mitochondria is even higher. However at the end there is an interesting result. I understand that there is a process to get to this point in the description of hormone-CcO interaction, but as due to size of text I lost the point indicated in the title. The inhibition of the generation of superoxide by T-hormone treated CcO looks more biologically plausible. Though, I must mention that data points for T3 is poor and for T4 due to narrow scale it is difficult to judge. Looking at the concentration of enzyme, which is 0.5uM and T4 Ki of 0.3uM it looks that the formed complex is extremely stable and provides almost stoichiometric inhibition. Could T4 be a cofactor or the CcO? Some explanation is provided in the discussion, but I think it should be more major part of it. Looking at the results and title I also wonder if BABS site can be described as biologically relevant binding site for thyroid hormones. Consider editing to showcase the unknown binding site.

Minor/editorial issues:

Some residual activity is expected for liposomes and mitochondria, as you discussed, but why there why there is residual activity in the isolated system?

As Fig 1 (or in discussion) I would propose to put a structure of the enzyme and proposed model of inhibition by T3&4.

The initial paragraphs of the discussion are the summary of authors previous study. Although relevant, I would not put it at the beginning, but rather through the text to explain the obtained results.

In all figures the numbers/text on the graphs are too small, please enlarge

Page 2, line 30-34 – this sentence is too-long.

Page 4 line 23 –  the paragraph formatting looks odd.

Page 4 line 24 – in the description of the method I’m missing the concentration of the CcO (should be added

Page 4 line 26 –  there are two spaces after TMPD. Also the next line beginning of the sentence. Also at line 42. There are numerous extra spaces in the text, please correct.

Page 6 – the Eq 1 should be a part of materials and methods as it is a standard equation

Page 6 line 37 –“ It is noteworthy…” I do not understand the sentence. According to fig 2B even at high concentration the activity remains above 25%, so it is not completely inhibited

Page 7 last paragraph – you write that inhibition is 50-60%. For the latter it would mean that 40% of activity remains. Looking in Fig 2 A & B the residual activity is 50 & 60%, respectively. Please correct

Round 2

Reviewer 1 Report

Good job done. All points are discussed. I accept.

Author Response

We really appreciate your positive feedback.

Reviewer 2 Report

First, I would like to thank authors for their effort to clarify and address my concerns, and I hope next review of previous comments and their replies help to improve the manuscript.

  • As regards the physiological implications of this work, I still find it difficult to show without further experiments despite explanations in the discussion session.  I understand this is a difficult issue but, as I remarked in my previous assessment, the partitioning explanation would be supported if Ki values decreased with the amount of non-polar phase in the protein samples because the increase in non-polar volume and the decrease of dimensionality within the membranes. I would expect then a lower Ki value for the vesicle-embedded complex, in contrast with authors’ data. 
  • Authors have addressed comment 2.
  • Regarding the inhibition of superoxide generation (comment 3) I understand the difficulties to measure in in mitochondria, as authors point out. Nevertheless, I wonder if it is still possible to measure it in vesicle—liposome—embedded protein samples, either directly or indirectly—profiting on spin traps and EPR or using reporter dyes—.
  • Authors have addressed comment 4 about the convenience of displaying docking results for comparison between steroids and thyroid hormones. Indeed, authors have docked T3, whereas in the previous report they mentioned (Buhrow et al. 2013), a molecule showing analogy with the hormone, not the hormone itself, was found to show similarity to steroids.

Reviewer 3 Report

The revised manuscript is greatly improved and all my concerns has been addressed. I recommend acceptance for publication.

Author Response

(The authors gave the same response as above.)
